# Applications and Future Prospects of Micro/Nanorobots Utilizing Diverse Biological Carriers

**DOI:** 10.3390/mi14050983

**Published:** 2023-04-29

**Authors:** Yu Lv, Ruochen Pu, Yining Tao, Xiyu Yang, Haoran Mu, Hongsheng Wang, Wei Sun

**Affiliations:** 1Department of Orthopedics, Shanghai Bone Tumor Institution, Shanghai General Hospital, Shanghai Jiao Tong University School of Medicine, Shanghai 200080, Chinayaphets.yang@sjtu.edu.cn (X.Y.);; 2College of Health Science and Technology, Shanghai Jiao Tong University School of Medicine, Shanghai 200025, China

**Keywords:** MNRs, bio hybrid

## Abstract

Targeted drug delivery using micro-nano robots (MNRs) is a rapidly advancing and promising field in biomedical research. MNRs enable precise delivery of drugs, addressing a wide range of healthcare needs. However, the application of MNRs in vivo is limited by power issues and specificity in different scenarios. Additionally, the controllability and biological safety of MNRs must be considered. To overcome these challenges, researchers have developed bio-hybrid micro-nano motors that offer improved accuracy, effectiveness, and safety for targeted therapies. These bio-hybrid micro-nano motors/robots (BMNRs) use a variety of biological carriers, blending the benefits of artificial materials with the unique features of different biological carriers to create tailored functions for specific needs. This review aims to give an overview of the current progress and application of MNRs with various biocarriers, while exploring the characteristics, advantages, and potential hurdles for future development of these bio-carrier MNRs.

## 1. Introduction

Recently, the rapid advancement of micromotors has led to significant progress in nanomedicine, particularly in targeted therapy for diagnostics and disease treatment [1,2,3]. Micro-nano robots (MNRs) move in a controlled manner by utilizing different external power sources like magnetic [4,5,6,7,8,9], optical [10,11,12], acoustic waves [13,14,15], heat [16], and other innovative methods [17]. MNRs have shown great potential in drug delivery [18], chemotherapy regimens [19], thrombus treatment [20], and other related applications. However, traditional MNRs face challenges, such as dynamic and manipulative problems in deep targeted drug delivery [21], high sensitivity and permeability requirements in complex conditions like the tumor microenvironment, tumor cell heterogeneity [22], and biocompatibility issues [23]. Additionally, MNR power and handling issues are concerning. For instance, chemical fuel-driven motors might not be suitable for microvascular applications, as exhaust gases can cause embolization in small blood vessels in the lungs or brain. As the micromotor travels deeper into the body, external power decreases, limiting its ability to move farther [24]. To overcome these challenges, researchers have turned to hybrid MNRs and bio-motors, which are highly biocompatible and specific [25].

Biohybrid micro-nano robots (BMNRs) combine artificial materials and biological entities, offering functions and advantages that only man-made motors lack [26]. Scientists have used bacteria [5,27,28,29,30,31], immune cells [32,33,34,35,36], and sperm [37,38,39,40,41] as common biological carriers, each having a certain degree of voluntary movement. Compared to previous physical models, most biohybrid MNRs integrate phagocytic ability and cell-specific characteristics, enhancing drug net charge and therapeutic effects. Special biological carriers, such as light-driven algae [42,43,44,45] and stem cells [46,47,48], are gaining attention, promising a bright future for BMNRs. This review summarizes different biological carrier categories supporting MNRs and their performance in their respective fields (Figure 1). It compares the relative merits between them and highlights the need for further research on specificity, power, and biosecurity. By comparing various biological carriers, the review showcases MNR applications and the potential for working with specific bio-carriers based on clinical use.

## 2. Nano/Micromotors Based on Immune Cells

When confronted with diseases of specific etiology, where conventional treatments are ineffective, MNRs have been predicted to be helpful to physicians with their good maneuverability and adaptability during treatment owing to some experiments conducted on animals [49], though further clinical study is still needed. The concept of cell-based MNRs was proposed in 2005 [50], and to date, cell-based MNRs can be broadly classified into somatic cells and germ cells. Somatic cells can be then simply divided into immune cells and non-immune cells. Immune cell carriers have unique properties that traditional MNRs lack, such as phagocytic capacity [51], specificity [52], chemotaxis [53], and good penetration when facing in vivo immune barriers like the blood–brain barrier. Research has focused on various immune cells, such as macrophages, natural killer (NK) cells, T and B lymphocytes, dendritic cells, and monocytes [54]. This review briefly discusses some of these cells and their application cases.

### 2.1. Nano/Micromotors Based on Macrophages

Macrophage-based MNRs possess unique characteristics not found in other immune cells. They have a strong inherent phagocytic capacity [55], which allows for a tight integration of immune cells and artificial machinery. Moreover, macrophages have a long half-life and excellent specificity, enabling increased drug payload during long-distance transport. The phenotype of macrophages can also be determined by the surface properties of the material they adhere to, allowing for functional changes under different stimulation conditions [56]. Macrophages are extensively used in tumor therapy because of their ability to home in on the tumor microenvironment through chemoattractive gradients [57]. However, commercial macrophages can sometimes trigger a host immune response, so using autoimmune cells as carriers might improve biocompatibility [58].

Combining a magnetic field drive with a macrophage drive is a classic model. Han et al. [32] designed a macrophage-based microrobot containing docetaxel-loaded nanoparticles for chemotherapy and magnetic nanoparticles for active targeting using an electromagnetic actuation system. In this study, tumor-associated macrophages were used as carriers to enhance drug concentrations. Macrophages were first transported to the target site under magnetic field control and then used their chemotactic function to penetrate tumor cells, driven by a mix of active movement and chemotaxis. The macrophage chemotaxis in the epithelial membrane antigen (EMA) system process shows promise for inhibiting tumor growth, but further in vivo and in vitro evaluations are needed. Apart from magnetic fields, near-infrared (NIR) laser induction is another major tool for targeting macrophages. Nguyen et al. [58] loaded macrophages with magnetic nanoparticles and doxorubicin (DOX) -containing thermosensitive nanoliposomes. This method avoided the direct toxicity of doxorubicin to macrophages and enabled magnetic targeting, while the photothermal effect under NIR radiation triggered drug release. Various in vitro experiments demonstrated the effectiveness of this approach. Cao et al. [35] considered the limitations of conventional manipulation modes and chose an acoustic manipulation system for macrophage-based MNRs containing magnetic nanoparticles. These motors can be pinpointed and rotated with high accuracy in a three-dimensional space. Acoustic manipulation offers high depth, good capture, and safety, with effectiveness supported by simulated experiments. However, its penetration requires further evaluation. Dai et al. [59] expanded on the use of macrophage carriers with MNRs, which can be precisely manipulated by magnetic fields (Figure 2a,b) and can also serve as a medium to manipulate other micro-motors, such as spermatozoa, through both contact and non-contact methods. Using a building block dipole-dipole structure, macrophage motors can form a chain-like cluster, greatly increasing their transport capacity, but potentially raising the risk of thrombosis.

However, despite significant progress in macrophage carrier motor research, some issues remain unaddressed. Toxicity in vivo still needs to be evaluated. In addition, the differentiation of macrophages should be further considered when they are used as BMNRs, which directly affects the therapeutic effect of the disease. It has been shown that MNRs themselves could affect macrophage differentiation. Song et al. [36] loaded positively charged DOX onto a magnetic microrobot by electrostatic action and made it exhibit pH-responsive release behavior. Notably, the surface of this magnetic robot was covered with nanoscale nickel and titanium to ensure a smooth surface, and the smooth surface of the nanorobot enabled macrophages to polarize in the M1 type. Macrophage differentiation is important in targeted tumor therapy, but the specific direction of macrophage differentiation should be assessed in typical macrophage BMNRs.

### 2.2. Nano/Micromotors Based on Other Immune Cells

Immune cells are diverse in type and function. For example, dendritic cells can be effective drug carriers [60], monocytes have potential to differentiate into dendritic cells and macrophages [61], neutrophils are abundant and first to reach inflammation sites [62,63], and well-studied T cells all hold promise in the bio-hybrid micro-nano-robotics field. Neutrophils, being highly abundant immune cells, make excellent biological carriers. Zhang et al. [64] encapsulated drug-loaded magnetic nano gel with *E. coli* and enabled neutrophils to engulf this bacterium, improving drug delivery efficiency and preventing premature drug leakage. External magnetic field manipulation allows accumulation in brain blood vessels, while autonomous neutrophil movement toward inflammation gradients enables crossing the blood–brain barrier to treat malignant gliomas.

In addition, NK cells and DC cells are made into BMNRs for active transport to the tumor sites. NK cells possess strong tumor-killing abilities and play a significant role as biological carriers in tumor-related fields. To boost NK cell virulence, Burga et al. [65] combined umbilical cord blood-derived NK cells with iron oxide nanoparticles (IONP), providing active magnetic-field-driven capabilities (Figure 2c). This improved NK cell cytotoxicity without affecting their phenotype and activity, suggesting magnetic driving has advantages over current methods. Furthermore, Song et al. [66] combined NK-92MI cells with sonazoid microbubbles using biotin/streptavidin conjugation, creating NK-Sona cells (Figure 2d), which can be imaged in real time using ultrasound. Sonazoid, a microbubble contrast agent, can infiltrate and enhance therapeutic target cells with ultrasound assistance, allowing for efficient drug delivery in the form of mRNA, antibodies, and immune cells [67]. The results show that sonazoid-coupled NK cells effectively kill tumors without losing efficiency. DC cells, the most functional antigen-presenting cells, are crucial for initiating, regulating, and maintaining immune responses, making them promising biocarriers. Cho et al. [68] synthesized multifunctional core-shell nanoparticles using magnetic NP (MNP) cores covered with a photonic ZnO shell. Internalized into DCs, these nanoparticles enable in vivo traceability and presentation of tumor antigens to DCs, resulting in strong anti- carcinoembryonic antigen (CEA) immune responses.

So far, there are still many cells that have not been exploited as BMNR, such as chimeric antigen receptor (CAR)-T cells, that could be well-suited to enhance immune cell infiltration and play a role in, for example, tumor therapy, if they could move in a targeted manner to the desired site. However, the uptake capacity of these cells is not as good as macrophages, making it difficult to introduce some magnetic particles into the cells. Stevens et al. [69] engineered two cells to adhere together by means of an engineered cell adhesion factor. Using this approach, it might be possible to link macrophages to cells such as CAR-T cells and achieve indirect transport of other immune cells through targeted transport of macrophages.

**Figure 2 micromachines-14-00983-f002:**
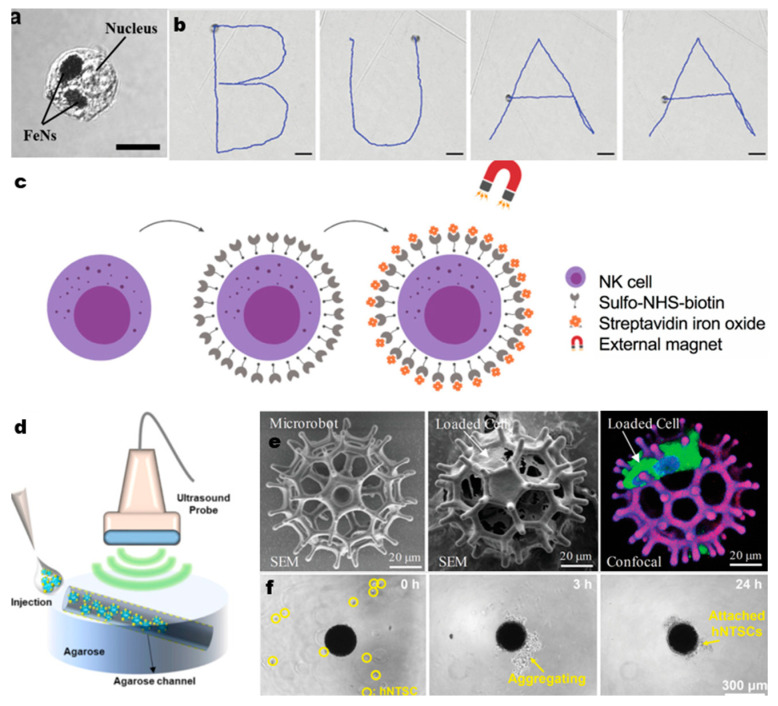
(**a**,**b**) Confocal image of the cell robot in which the black part indicates the internalized iron nanoparticles (FeNs), scale bar: 10 µm. Trajectories of a cell robot in the rolling mode moving along a predefined “BUAA” shaped track. Scale bar: 40 µm. Reprinted/adapted with permission from Ref. [59]. 2022, Wiley-VCH GmbH. (**c**) Schematic of the bioconjugation strategy to generate the NK cell–iron oxide nanoparticle biohybrid (NK:IONP). Reprinted/adapted with permission from Ref. [70]. 2019, American Chemical Society. (**d**) Schematic diagram of the agarose flow phantom model and ultrasound probe. The wall-less tube was 1 mm in diameter and slanted at an angle of 11.31°. Reprinted/adapted with permission from Ref. [65]. 2021, Burga. (**e**) SEM images of microrobot structure (left), cell-loaded microrobot (middle), and confocal scan of GFP-MSCs cultured on the microrobot (right). In the confocal image, cell nuclei were stained with Hoechst 33342, and the microrobot was in purple. Reprinted/adapted with permission from Ref. [71]. 2020, Wiley-VCH GmbH. (**f**) Time-lapse bright-field images of 3D cell culture for the formation of a spheroid with the GelMA microrobot. Reprinted/adapted with permission from Ref. [33]. 2020, Wiley-VCH GmbH.

## 3. Nano/Micromotors Based on Non-Immune Cells

Blood circulation, lymphatic circulation, and released cytokines drive immune cells to reach certain places as BMNRs for transport or functional purposes. In blood circulation, red blood cells could also act as BMNRs, but red blood cells could only function as carriers for no other organelles in them. Other cells such as stem cells or fibroblasts, which have a weak migratory capacity compared to immune cells but play an important role in biological processes such as tissue repair and tumorigenesis, can also serve as BMNRs [72].

### 3.1. Nano/Micromotors Based on Stem Cells

Mesenchymal stem cells (MSCs) can differentiate into various mature cells under specific conditions. Stem cells have great potential, but the challenge is how to strictly direct and manipulate their differentiation. MNRs offer a possible solution for controlled stem cell differentiation [73]. Current research mainly involves nanoparticles such as upconversion nanoparticles (UCNPs), quantum droplets (QDs), MNPs, mesoporous silica nanoparticles (MSNs), graphene oxide (GO), plasmonic gold nanoparticles (AuNPs), and carbon nanobelts (CNBs) [73].

Mesenchymal stem cells are widely studied in the field of micro and nano robotics. Go et al. [74] designed a magnetic cellular micro-scaffold composed of poly(lactic-co-glycolic acid) (PLGA) for targeted MSC delivery for articular cartilage regeneration. The framework is coated with amine-functionalized MNPs, enabling targeted stem cell transport and manipulation by an external magnetic field. However, the issue of stem cell differentiation has not been described in detail. Kang et al. [75] designed an upconversion nanotransducer-based nanocomplex (UCNP) with photolabile caging of chondro-inductive kartogenin (KGN). This structure allows UCNP to be used as a pathway for light-induced stem cell differentiation. In another study by Go et al. [76], a similar magnetic microscaffold is immobilized with TGF-β1, allowing MSCs to ensure stable differentiation even in disease settings with minimal secretion of bioactive molecules. Wei et al. [71] designed a magnet-driven, image-guided degradable microrobot to deliver MSCs to treat hepatocellular carcinoma in mice. This robot has a burr-like porous spherical structure, providing biodegradability, mechanical strength, and magnetic drive capability (Figure 3a). The study also designed a photoacoustic imaging technology that combines acoustic imaging’s depth with optical imaging’s high resolution. This allows precise, real-time positioning of the robot by photoacoustic tomography (PAT) within 2 cm of the body. Nasal turbinate stem cells are also good vehicles for modification. Jeon et al. [48] designed a magnetically driven micro-nano-robot called Cellbot by internalizing MNPs into human nasal turbinate stem cells for minimally invasive brain delivery. The final Cellbots can proliferate and differentiate into neurons, neural precursor cells, and glial cells in the brain’s rich environment of bioactive molecules.

Otherwise, fibroblasts are often selected for their repair capacity. Gyak et al. [77] developed a biocompatible, magnetically activated silicon carbonitride (SiCN) ceramic microrobot, loaded onto fibroblasts for wound-healing applications. This ceramic robot has good mechanical stiffness, biocompatibility, and actuation ability under magnetic field manipulation. To address clinical and patient needs, Noh et al. [33] suggested a spherical gelatin methacrylate (GelMA)-based micro-nano-robot to carry SPION and deliver human nasal turbinate stem cells. They considered the challenges in manufacturing MNRs and biodegradation speed, comparing it with materials like poly(ethylene glycol) diacrylate (PEGDA), and poly(lactic-co-glycolic acid) (PLGA). The robot is created using a flow-focusing droplet generator process, which is fast and precise, allowing for complete enzymatic degradation by collagenase, ultimately enabling stem cells to differentiate into neuronal cells. Therefore, stem cells are usually not used as drug carriers, but rather as a vital part of therapy itself. It is crucial to ensure controlled induction and maintain biosafety when working with stem cell vectors.

### 3.2. Nano/Micromotors Based on Red Blood Cells

Red blood cells have unique surface properties [78,79,80] and serve as long-circulating delivery vehicles, making them excellent carriers in the blood. Wu et al. [81] reported a biomimetic motor sponge, which combines a biocompatible gold nanowire motor with erythrocyte nanoscale properties. Driven by an ultrasonic field, it absorbs toxins that disrupt cell membranes, thus completing bio-detoxification. Wu et al. [82] introduced MNPs asymmetrically into erythrocytes, using their inherent asymmetric shape and distribution for ultrasonic actuation, while using an external magnetic field for fine directional control and guidance, thus achieving drug delivery. Red blood cells are not only powerful carriers but also a raw material for the preparation of pure cell membranes. Cell membranes can be used to wrap nanoparticles to form bionanocytes [8,83]. However, there are many challenges that need to be overcome, such as their fragile membranes and limited drug-carrying capacity.

## 4. Nano/Micromotors Based on Sperm Cells

Sperm cells can also be used as biological carriers in the human body. Mammalian sperm cells have strong autonomous propulsion [39], chemotaxis [84], and versatility when used as biological carriers or bionic robotic templates. However, research on sperm bio-motors faces challenges, such as the presence of anti-sperm antibodies (ASAs) and the size and rigidity of bionic manufactured motors [85]. The transverse waves [86] generated during sperm motility also offer the possibility of controlled motility [38]. However, research on sperm bio-motors also faces many problems. The larger size [87] and rigid shell [88] of bionic manufactured motors can be detrimental to the organism, despite their rapid surging speed. In contrast, flexible motors that are covered with too many nanoparticles can have reduced motility [41].

To solve this problem, Chen et al. [40] protected biohybrid sperm motors from their surroundings during drug delivery via multifunctional metal–organic framework exoskeletons. They fabricated metal organic frameworks (MOFs) and zeolitic imidazolium framework-8 (ZIF-8) nanoparticles (NPs) for encapsulating sperm cells via the complexation with tannic acid (Figure 3a). Such binding has little effect on the dynamics of the sperm themselves and protects the sperm cells from ASAs damage through selective penetration and oxidation through the continuous release of zinc ions. This design leaves the source of power entirely to the sperm themselves and instead mechanizes the sperm cells by protecting the sperm from the environment.

Based on not compromising motility, some researchers began to pursue the manipulability of sperm motors. Magdanz et al. [89] reported a magnetic microtubule consisting of ferromagnetic layers for holding sperm. The diameter of this microtubule is slightly larger than the head of the sperm, allowing it to be mechanically trapped once the sperm enters. By varying the temperature, the speed of sperm movement can be manipulated by taking advantage of the sperm’s tendency to heat, thus achieving the function of stopping and advancing. The magnetic microtubules also give the sperm the ability to be finely manipulated by an external magnetic field. This process reduces the speed of sperm movement by approximately 10% compared to their natural state. This research not only contributes to reproductive medicine but also offers the potential for precision medicine and targeted drug delivery.

In reproductive medicine, it is essential to maintain sperm vitality. Medina-Sanchez et al. [90] produced microhelices to transport motility-deficient sperm using a rotating magnetic field for propulsion (Figure 3b). This research not only contributes to reproductive medicine but also has potential for precision medicine and targeted drug delivery. There are still challenges to address, such as potential damage to sperm cells by robotic structures, biodegradation of remaining material after transport, and maintaining sperm cell viability.

**Figure 3 micromachines-14-00983-f003:**
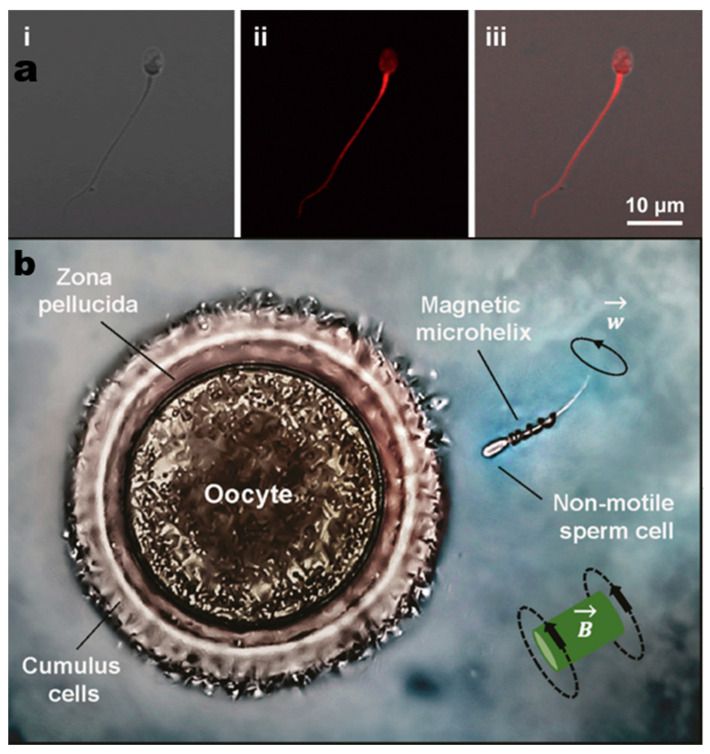
(**a**) Fluorescence images of ZIF Spermbots with RhB-labeled ZIF-8 NPs: (**i**) optical channel, (**ii**) RhB channel, and (**iii**) merged image of optical and RhB channels. Reprinted (adapted) with permission from Ref. [40]. 2021, American Chemical Society. (**b**) An immotile sperm is captured by a remotely controlled magnetic helix and delivered to the oocyte for fertilization. Reprinted/adapted with permission from Ref. [90]. 2016, American Chemical Society.

## 5. Nano/Micromotors Based on Microbe

Compared to cell-based MNRs, microbe-carrier MNRs have very many unique properties. In contrast to most cell-based motors that require active actuation, the most important feature of microbe-based motors is their sensing and self-driving capabilities [91]. Especially, they have good performance in the face of low Reynolds number environments. This is due in large part to the transverse wave that they generate during their movements [86]. This property of converting chemical energy into flagellar-driven mechanical energy allows microbial-based MNRs to be designed with only the manipulation in mind and without the need to provide an additional power source. The most commonly used microbial-based MNRs today include bacterial and algal carriers.

### 5.1. Nano/Micromotors Based on Bacteria

Bacterial carriers were first used in microfluidic studies [43]. Back in 2006, Tung et al. [92] immobilized *E. coli* on the inner surface of a microfluidic chip and relied on the rotation of the bacteria to pump the fluid in the microfluidic channel. Based on the bacteria’s properties, the BMNRs have unusual characteristics. The bacteria are easily modified by engineered plasmids and are highly manipulable, and some bacteria have the good penetrating ability and anti-tumor activity in the face of tumor therapy [93]. Bacteria can also respond to a variety of factors in the host environment, including temperature, oxygen, and pH, and thus have a certain degree of chemotaxis [31],which allows bacteria to have a very important role in anti-infection and tumor therapy. At the same time, bacteria are inherently immunogenic, which allows them to further stimulate the body’s immunity when delivering drugs to environments with low immunogenicities, such as TME [94]. The combination of these factors makes bacterial vector bio-motors primarily relevant in the field of oncology. The sensitivity of bacteria to antibiotics also ensures their biosafety to a certain extent.

Engineered bacteria and bacterial bionic materials are widely used in BMNRs. Rogowski LW et al. [95,96] have developed a flagellar-propulsion-based nanoparticle, mimicking the flagella of *Salmonella typhimurium*. Cheng et al. [30] designed a composite bacterial robot. This robot is simultaneously sensitive to magnetic, thermal, and hypoxic environments and can improve the reporting of heat and location signals in targeted cancer therapies through internal fluorescent proteins. The perception of magnetism and hypoxia comes from *E. coli* Nissle 1917 loaded with MNPs, while the reporting of heat and position comes from a thermal-logic circuit loaded within the bacteria. The EcN gene is encoded with the NDH-2 or NDH-2/mCherry gene, which functions to induce reactive oxygen species as a means of anti-tumor therapy. Beyond oncology treatment, bacterial tropism can also be used in oncology diagnosis. Park et al. [27] combined attenuated S. typhimurium, which has high athleticism with Cy5.5-coated polystyrene microbeads through the high-affinity interaction between biotin and streptavidin, and performed bacterial detection and localization by arterial luciferase (lux) or green fluorescent protein (GFP) expressed by the bacteria. These bacteria were sensitive to environmental stimuli and their biosafety was ensured because of the attenuated toxicity. However, the low drug-carrying concentration and the limited simulated environment still left several questions about the study.

In biological environments distant from human society, certain rare bacteria deserve our attention. For example, Song et al. [29] developed a bacterial robot using magnetotactic cells with powerful motility. The *MO-1 bacterium*, a type of polar magnetotactic bacteria, can move along magnetic lines of force with the help of its two sturdy flagella. Each flagellum consists of seven filaments enclosed in a specialized sheath and can travel at speeds up to 300 μm/s. The researchers bound the bacteria to polystyrene microbeads using an antigen-antibody reaction, creating a firm bond between them. In experiments, the bacteria swam effortlessly in microfluidic channels, guided by a rotating magnetic field. They also functioned as stirrers and demonstrated potential for detecting and capturing pathogens. Meanwhile, Schürle et al. [28] utilized a naturally magnetic bacterium from the *Magneto spirillum* genus, which contains iron oxide particles. By applying an external rotating magnetic field, they directed the bacterium through the vessel wall near cancer cells. The size of the cell gap in the vessel wall was temporarily adjusted to allow the bacteria to pass through. Notably, rotating magnetic fields offer several advantages over static magnetic fields. The rotating magnetic field is highly propulsive and does not rely on the active movement of bacteria before entering the tumor microenvironment, increasing efficiency. In addition, rotational motion along the vessel wall increases the chances of bacteria entering the vessel wall and reduces off-targeting. Once in the vicinity of the lesion, the bacteria can then be targeted by their chemotaxis to reach the central part of the tumor. However, the team only verified the effect of the magnetic field on bacterial aggregation at the target cells but did not make the bacteria carry the drug, so the clinical efficacy needs to be further investigated.

For tumor therapy, bacterial robots are equally effective. Park et al. [97] combined a paclitaxel-loaded liposomal microcargo with tumor-targeting *Salmonella typhimurium* bacteria. This bacterially driven liposome has a higher mobility compared to regular liposomes and demonstrated a strong tumor-killing ability in an in vitro test based on a breast cancer cell line (4T1), heralding a promising future for a nanomotor based on bacteria in tumor therapy.

Despite their potential as biological vectors, the use of bacteria in this capacity carries several risks. Firstly, due to their pathogenic nature, many bacteria can be rapidly eliminated from the bloodstream by the immune system. Second, achieving effective attachment of bacteria to micro-nanostructures is a challenging task. Lastly, bacteria possess limited propulsive force, which can result in poor targeting and movement of the MNRs. Currently, *Listeria*, *Escherichia*, *Clostridium*, and *Salmonella* are the most studied bacteria for cancer gene therapy [98,99]; whereas the specificity of bacterial motors can be controlled by promoters, and specific promoters that are active only when induced by specific factors are most commonly used [100].

### 5.2. Nano/Micromotors Based on Algae

Algae have also been utilized as carriers for MNRs. Algal cells can convert light energy into mechanical energy through the process of photosynthesis [101], and their flagella-driven movement allows them to be used as self-propelling MNRs. Among the different types of algae, *Chlamydomonas reinhardtii* has been widely studied for its potential in MNR applications.

One of the main advantages of using algae-based MNRs is their ability to be powered by light, which is a non-invasive and renewable energy source. This property has been exploited in the development of light-driven algal micromotors. For instance, Xie et al. [43] reported the fabrication of an algal micromotor using *Chlamydomonas reinhardtii* and demonstrated its ability to transport cargo under light irradiation. The algal micromotor was able to propel itself in response to light and deliver the cargo to a desired location. Another advantage of algae-based MNRs is their biocompatibility and biodegradability [43], which makes them a promising option for drug delivery and other biomedical applications. Algae can be easily modified genetically or chemically, and their surface can be functionalized with different targeting molecules or therapeutic agents.

However, there are still several challenges associated with the use of algae-based MNRs. One of the main issues is the limited propulsion force generated by the algae, which can result in inefficient transport of the MNR and its cargo. Additionally, the sensitivity of algae to environmental factors, such as temperature and pH, can affect their viability and functionality. Moreover, the potential immune response elicited by the algal cells needs to be considered when designing algae-based MNRs for biomedical applications. However, not all algae are phototropic, and thus, magnetic nano-beads, which are commonly used in other bio-carrier MNRs, come in handy. Liu et al. [102] proposed a biohybrid magnetized microrobot based on *Thalassiosira weissflogii* frustules to which MNPs are attached by electrostatic adsorption. Algae has a natural porous silica structure, a large surface area, is stable and heat resistant, and is a good carrier for drugs. The diatoms themselves have poor motility, whereas the MNRs designed by this method have high drug-loading capacity, controlled and flexible motion, the ability to switch between two different modes of motion, and the ability to release drugs based on pH sensitivity. By loading doxorubicin, targeted therapy against MCF-7 human breast cancer cells was achieved.

Zhang et al. [44] loaded magnetized nanomaterials onto *Spirulina* by a sol-gel process and loaded it with DOX. The micro-robot not only has an efficient propulsion performance with a maximum speed of 526.2 μm/s under a rotating magnetic field but also has a pH+NIR dual manipulation drug release mechanism with a high drug-loading capacity and a wide range of drug release means. Under the action of a magnetic field, algae can be integrated into clusters. Cai et al. [45] covered MNPs with *Chlorella* and simultaneously loaded DOX. By using magnetic dipolar interactions, the robotic units undergo reversible assembly, reconfiguring into chain-like motors as tiny dimers and trimers. Such aggregates can be rolled and tumbled and can reach speeds of 107.6 μm/s under the application of a magnetic field. The algal motors are highly drug loaded, flexible, simple to manufacture, and pH-sensitive and have also been shown to be tumor-killing in in vitro experiments on Hela cells.

Microbe-based MNRs, including those based on bacteria and algae, offer unique properties and advantages for various applications, such as drug delivery, diagnosis, and environmental sensing. However, there are still several challenges to overcome, such as improving the propulsion force, enhancing targeting capabilities, and ensuring biocompatibility and biosafety. From the existing research on microbial vectors, it is clear that the prevention of immune responses triggered by microbial antigenicity is crucial. Further research and development in this field are needed to optimize microbe-based MNRs and unlock their full potential for a wide range of applications.

## 6. Summary and Expectation

In summary, bio-carrier MNRs have shown great potential for various therapeutic applications, especially targeted drug delivery for tumors. The unique autonomous biological functions of bio-carriers provide these MNRs with exceptional maneuverability and autonomous actuation capabilities, allowing precise drug delivery to otherwise inaccessible lesions. The bio-derived structure also significantly improves the efficiency and safety of these MNRs. However, the stability and practicality of bio-hybrid MNRs in complex human environments still need further exploration and optimization.

A competent bio-carrier micro/nanorobot of the future should possess the following characteristics: (A) good biocompatibility, avoiding recognition as a foreign body and clearance by the immune system, (B) high carrying capacity to improve transport efficiency, (C) high carrying capacity to improve transport efficiency, (D) high maneuverability, with both active manipulation and spontaneous drive modes, (E) good biosafety, allowing for simple clearance after task completion, and (F) easy and cost-effective manufacturing. Building on these characteristics, researchers should also consider the natural properties of different biological carriers to develop specialized motors adapted to various situations.

As research progresses, bio-hybrid MNRs are expected to become increasingly versatile and better suited for diverse applications. Continued innovation and development in this field will unlock the full potential of bio-carrier MNRs for targeted drug delivery, diagnostics, environmental sensing, and other important applications.

## Figures and Tables

**Figure 1 micromachines-14-00983-f001:**
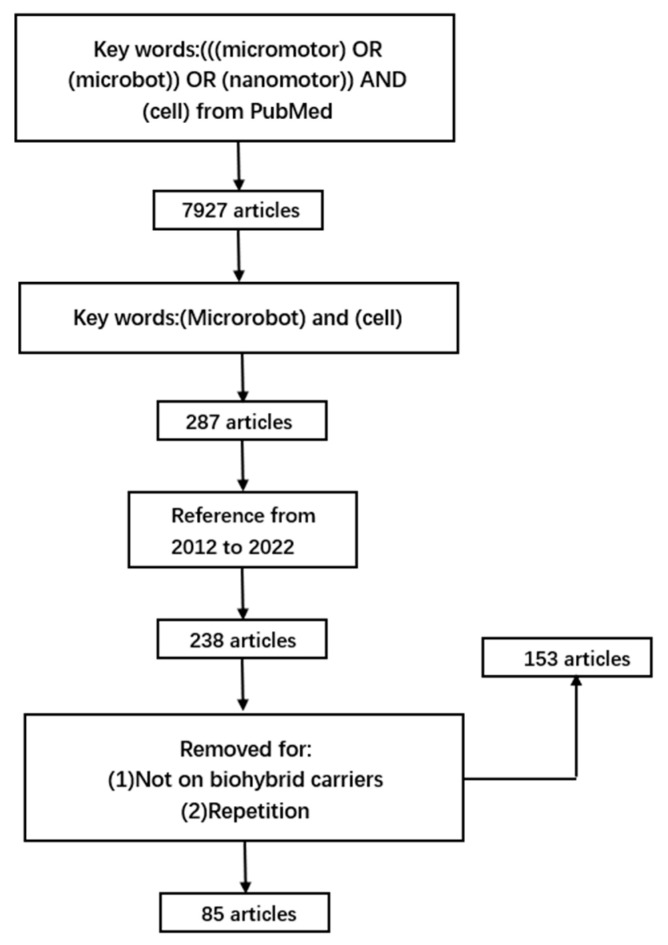
Review strategy.

## Data Availability

No new data were created or analyzed in this study. Data sharing is not applicable to this article.

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
