# Peer review of "Applications and Future Prospects of Micro/Nanorobots Utilizing Diverse Biological Carriers"

_micromachines, 2023, doi:10.3390/mi14050983_

Round 1
Reviewer 1 Report
This paper mainly discussed the current works about the micro/nanorobot with various biological carriers, including immune cells, non-immune cells, sperm, and microbe. Overall, this review paper provides an overview of the current works and technologies about the biocarriers for different kinds of bio-applications of the micro/nanorobot. My comments are as following:
1. Firstly, in the introduction part, “Recently, owing to the rapid development of microrobots, nanomedicine has made huge progress in the diagnostics and treatment of disease, especially in targeted therapy.[1-3]”. This statement has no problems, but I suggest the authors to cite more recently review papers to fully support this part of statement. I found these three references were published more than 10 years ago. There are plenty of new works have been reported within these years containing more exciting and attractive works.
2. And the references [4-6] were cited to illustrate the magnetic field power source. As the authors may know, there are kinds of magnetic field which can be used to actuate and control the micro/nanorobots. But I found reference [5-6] are not strong enough to represent the current manipulation method of the magnetic field. I highly recommend the authors search more powerful papers to fully support the statement about the magnetic field. As I know, magnetic field manipulation has kinds of operation mode, like the following:
1) rotating magnetic field
i. Peyer, Kathrin E., Li Zhang, and Bradley J. Nelson. "Bio-inspired magnetic swimming microrobots for biomedical applications." Nanoscale 5, no. 4 (2013): 1259-1272.
ii. Servant, F. Qiu, M. Mazza, K. Kostarelos and B. J. Nelson, "Controlled in vivo Swimming of a swarm of bacteria-like microrobotic flagella", Adv. Mater., vol. 27, pp. 2981-2988, 2015.
2) gradient magnetic field
i. E. Diller, J. Giltinan and M. Sitti, "Independent control of multiple magnetic microrobots in three dimensions", Int. J. Robot. Res., vol. 32, pp. 614-631, 2013.
ii. Li. D, Niu F, Li J, et al. Gradient-enhanced electromagnetic actuation system with a new core shape design for microrobot manipulation[J]. IEEE Transactions on Industrial Electronics, 2019, 67(6): 4700-4710.
3) alternating magnetic fields
i. Xie, H., Sun, M., Fan, X., Lin, Z., Chen, W., Wang, L., ... & He, Q. (2019). Reconfigurable magnetic microrobot swarm: Multimode transformation, locomotion, and manipulation. Science robotics, 4(28), eaav8006.
ii. Xing, Liuxi, et al. "A New Drive System for Microagent Control in Targeted Therapy Based on Rotating Gradient Magnetic Fields." Advanced Intelligent Systems 4.9 (2022): 2100214.
3. As the authors mentioned kinds of manipulation method of the micro/nanorobots, like the magnetic field, optical, acoustic wave, heat, etc. It is better to use a figure showing all these kinds of manipulation methods.
4. In the “Nano/Micromotors based on Macrophage” part, the authors mentioned several points of manipulability of macrophage carrier robots that remain unresolved, except these points, the pass ability and toxicity of the micro/nan robots in the in vivo environments should also listed as the consideration.
5. I am wondering why the authors mentioned the micro/nanomotors in some of the parts and the micro/nanorobots in some other parts. Were the authors trying to describe two different kinds of micro/nanostructures? If so, I suggest the authors clarify the difference between these two kinds of structures. If not, maybe it is better to only use one word to avoid confusing the readers.
6. Except the mentioned various biological carriers, as I know, there are also some works have been reported which use the cell membrane as the biological carriers. I am not sure if the authors think the cell membrane can be considered as one kind of biological carriers. I know it depends on whether the carrier is a living organism or not. But if the authors think the cell membrane can be considered as a biological carrier, please see the following references:
1) Zhang, Fangyu, et al. "Nanoparticle-modified microrobots for in vivo antibiotic delivery to treat acute bacterial pneumonia." Nature Materials 21.11 (2022): 1324-1332.
2) Fang, Ronnie H., Weiwei Gao, and Liangfang Zhang. "Targeting drugs to tumours using cell membrane-coated nanoparticles." Nature Reviews Clinical Oncology 20.1 (2023): 33-48.

Author Response
Answer to Reviewer 1
Thanks for your suggestions and here are our answers.
- Firstly, in the introduction part, “Recently, owing to the rapid development of microrobots, nanomedicine has made huge progress in the diagnostics and treatment of disease, especially in targeted therapy.[1-3]”. This statement has no problems, but I suggest the authors to cite more recent review papers to fully support this part of the statement. I found these three references were published more than 10 years ago. There are plenty of new works have been reported within these years containing more exciting and attractive works.
Answer: Thanks for your suggestion. And we have updated the reference in line 28-29 yellow labeled.
- And the references [4-6] were cited to illustrate the magnetic field power source. As the authors may know, there are kinds of magnetic field which can be used to actuate and control the micro/nanorobots. But I found reference [5-6] are not strong enough to represent the current manipulation method of the magnetic field. I highly recommend the authors search more powerful papers to fully support the statement about the magnetic field. As I know, magnetic field manipulation has kinds of operation mode, like the following:
Answer: Thanks for your suggestion. And we have updated the reference in line 30 yellow labeled.
- As the authors mentioned kinds of manipulation method of the micro/nanorobots, like the magnetic field, optical, acoustic wave, heat, etc. It is better to use a figure showing all these kinds of manipulation methods.
Answer: Thanks for your suggestion. We have reduced the number of images, considering that energy is not our topic, and we have only elaborated in the introduction.
- In the “Nano/Micromotors based on Macrophage” part, the authors mentioned several points of manipulability of macrophage carrier robots that remain unresolved, except these points, the pass ability and toxicity of the micro/nan robots in the in vivo environments should also listed as the consideration.
Answer: Thanks for your suggestion. We have added in line110 yellow labeled.
- I am wondering why the authors mentioned the micro/nanomotors in some of the parts and the micro/nanorobots in some other parts. Were the authors trying to describe two different kinds of micro/nanostructures? If so, I suggest the authors clarify the difference between these two kinds of structures. If not, maybe it is better to only use one word to avoid confusing the readers.
Answer: Thanks for your suggestion. We have a unified statement using micro/nanorobots/MNRs.
- Except the mentioned various biological carriers, as I know, there are also some works have been reported which use the cell membrane as the biological carriers. I am not sure if the authors think the cell membrane can be considered as one kind of biological carriers. I know it depends on whether the carrier is a living organism or not. But if the authors think the cell membrane can be considered as a biological carrier, please see the following references:
Answer: Thanks for your suggestion. A cell membrane is indeed a tool that can be used as a bionic cell, and we add this paragraph to the description of " Nano/Micromotors based on red blood cells" because red blood cells do not contain cell lines, and it is possible to prepare a relatively pure cell membrane.

Reviewer 2 Report
The entire manuscript needs extensive English editing; it seems that the manuscript has not been proofread. In addition to the ambiguity in multiple places of the text, too many grammatical mistakes and the use of the awkward word and sentences can be found in the entire manuscript. On many occasions, very long sentences (sometimes more than 100 words in a sentence) make the manuscript very difficult to read and comprehend. In several cases, the paragraphs are not discussing and following any specific idea and seem to be only a repetition of the abstract of the corresponding studies (references). At the end of each subsection, the authors try to mention the challenges and the issues of each cell type, but it is not clear how they reached those conclusions? Because almost there is no clear discussion about these issues and no reason or references for these problems that are presented. This can be one of the main problems of the manuscript, as it seems to be only rephrasing the abstract of the considered studies in many cases. In addition, the motivation of the review is not clearly and well described. It is also better to clearly mention the outline and the scope of the review in the introduction. It is not possible for the reader to understand what is specifically going to be discussed in the review unless going through the main text. Moreover, the database and methodology that the authors have used for choosing the selected studies are totally absent.
Finally, there are too many general or misleading statements, mostly in the introduction, which either are hard to understand or no clear conclusion can be drawn from them. Moreover, several acronyms have been used in the manuscript which has never been defined. Here are only a few of these problems as examples, but they're more:
- Three figures have been adapted and no description of them is presented in the main text.
- In the abstract MNR has not been defined.
- What do the authors mean by fuel problem in the abstract? It is a very general statement.
- What is the difference between “micro-nano robot”, and “micro/nanomotor”, if they are the same or different, it should be clarified. In addition, “Nano/micromotor” has also been used. A unified term is better to be used throughout the manuscript.
- Line 43, what does “certain self-function without exception” mean?
- What are “power sources”?
- Line 31, “micro/nanomotors have performed so well that some 31 previous medical procedures are almost replaced”, have authors provide any study proving this clinically?
- If a Biohybrid nanomotor is a mixture of artificial materials and biological entities, it is also a man-made robot.
- Line 51, “foreboding the spacious prospect of Biohybrid nanomotor”.
- Line 55-56, “As a result, by comparing various biological carriers, we indicated 55 the possibility to work on another potential carrier.” Is it a motivation for writing the review? Do you mean introducing new carriers?
- Line 103. “The effect of the EMA system on tumor cell survival is effective”, Is EMA beneficial for the survival of tumors?!!!
- Line 105, “NIR” has not been defined.
- The authors should decide to use either superparamagnetic, superparamagnetic, SPIONs, magnetic nanoparticles, or MNPs. Throughout the manuscript, these terms have been used when describing superparamagnetic iron oxide nanoparticles.
- What is “MÏ•s” in line 169?
- Paragraph in lines 114 to 123 is inaccurate, the work by Cao et al. (ref#32) is not optically based nor they have examined any other methods except the acoustic method in their study.
- In lines 124-134, the paragraph describing the work by Dai et al. (ref#56) is also very confusing and inaccurate. For example, what is “building block dipole-dipole structure dipole-dipole”? Also, the authors mentioned that “macrophage robots can be built into a chain-like cluster more than a hundred times their original size”, while in ref32, the study shows that the cell robots can also form chain-like swarms to transport a large object (more than 100 times the volume).
- Line 232, UCNPs, QDs, MSNs, GO, AuNPs, and CNBs have not been defined.
- What is “micmicro scaffoldmposed” in line 237?
- What are MSCs in line 238?
- What is hNTSC in line 271?
- PA line 284?
- The sentence in line 307 has no verb!
- ….
Author Response
Answer to Reviewer 2
Thank you very much for your suggestions, we are sorry for the problems and have made serious corrections to them
- We have embellished the language and abbreviations.
- We have added the figures in the main text.
- MNR in the abstract has been defined and be used in the whole text.
- For the question “What do the authors mean by fuel problem in the abstract?” Some chemically driven fuels limit their action in the blood system or body fluids due to the release of gases or other substances. We have restated our ideas in the statements in line 15-17 yellow labeled.
- “The macrophage chemotaxis in the epithelial membrane antigen (EMA) system process shows promise for inhibiting tumor growth”, we have reconfirmed the EMA is not beneficial for survival of tumors. There was a problem with our presentation and we are changing this in line 90-92 yellow labeled, sorry.
- We have using IONP in line 136 and 166 to describe iron oxide nanoparticle.

Reviewer 3 Report
This review work provides an extensive discussion of the micro/nanorobots research for various therapeutic needs with the specialization of targeted drug delivery for tumors. It provides summaries of the advantages for those micro/nano robots compared to conventional ones, the referenced works are overall representative and up to date. I my view, this review is qualified for acceptance after some minor comments below are addressed.
1. All the abbreviations should be added at the place where the terminologies are firstly shown, such as MNR and BMNRs in the abstract and all the others from the referenced articles for readability.
2. In the introduction section, the authors mentioned the approaches for controlling micro/nano robots, but the references seem to be not representative enough as there are quite several excellent progresses being done by different research groups across the world, such as Dr. Yu Sun’s group in University of Toronto, Dr. Bradley Nelson’s group in ETH Zurich, Dr. Li Zhang’s group in the Chinese University of Hong Kong, Dr. Peer Fischer’s and Dr. Metin Sitti’s groups in Max Planck Institute, etc. I would suggest the authors to provide more discussions to solidify the content of this part as those works contribute quite a lot in the development of micro/nano robot for medical purpose. Also, there are already several companies working on commercializing this kind of applications, such as Bionaut Labs.
3. In section 2.1, the authors discussed several developments of micro/nano robot for in vivo applications, such as control based on magnetic optical, and acoustical methods. One of the current limitations that the workspace usually is limited due to the physical principles or hardware limitations, such as power output or safety issues, in the control, while the main objective for those applications is to get the micro/nano robots controlled within the human body. I’m wondering if the authors can provide some discussion about this part or provide some related work?
4. In section 4, the authors discussed the helical attachment research for improving the vitality in the micro/nano robot applications, there has been similar research work conducted by other research groups before the referenced articles, such as work from Dr. Min Jun Kim’s group in Southern Methodist University that uses repolymerized flagellum to provide extra propulsion. I’m wondering if the authors can provide more discussion to make this section more extensive as a review work.
Author Response
Answer to Reviewer 3
Thank you very much for your suggestions, we are sorry for the problems and have made serious corrections to them
- All the abbreviations should be added at the place where the terminologies are firstly shown, such as MNR and BMNRs in the abstract and all the others from the referenced articles for readability.
Answers: Thanks for your suggestion. And we have corrected it.
- In the introduction section, the authors mentioned the approaches for controlling micro/nano robots, but the references seem to be not representative enough as there are quite several excellent progresses being done by different research groups across the world, such as Dr. Yu Sun’s group in University of Toronto, Dr. Bradley Nelson’s group in ETH Zurich, Dr. Li Zhang’s group in the Chinese University of Hong Kong, Dr. Peer Fischer’s and Dr. Metin Sitti’s groups in Max Planck Institute, etc. I would suggest the authors to provide more discussions to solidify the content of this part as those works contribute quite a lot in the development of micro/nano robot for medical purpose. Also, there are already several companies working on commercializing this kind of applications, such as Bionaut Labs.
Answers: Thanks for your suggestion. And we have added them and references at line 30-38.
- In section 2.1, the authors discussed several developments of micro/nano robot for in vivo applications, such as control based on magnetic optical, and acoustical methods. One of the current limitations that the workspace usually is limited due to the physical principles or hardware limitations, such as power output or safety issues, in the control, while the main objective for those applications is to get the micro/nano robots controlled within the human body. I’m wondering if the authors can provide some discussion about this part or provide some related work?
Answers: Thanks for your suggestion. We have rewritten the section 2.1.
- In section 4, the authors discussed the helical attachment research for improving the vitality in the micro/nano robot applications, there has been similar research work conducted by other research groups before the referenced articles, such as work from Dr. Min Jun Kim’s group in Southern Methodist University that uses repolymerized flagellum to provide extra propulsion. I’m wondering if the authors can provide more discussion to make this section more extensive as a review work.
Answers: Thanks for your suggestion. And we have added the works of Dr. Min Jun Kim’ group in “Nano/Micromotors based on bacteria” line 314-316.
Round 2
Reviewer 1 Report
All the questions have been well addressed. The quality of this paper has been improved after modify.
Author Response
Thanks for your review.
Reviewer 2 Report
authors incorporated successfully the reviewer's comments. It can be published in a current form
Author Response
Thanks for your review.